# Phage-assisted evolution of highly active cytosine base editors with enhanced selectivity and minimal sequence context preference

Emily Zhang [1,2,3], Monica E. Neugebauer[1,2,3], Nicholas A. Krasnow[1,2,3] & David R. Liu [1,2,3] ✉

TadA-derived cytosine base editors (TadCBEs) enable programmable C•G-to-T•A editing while retaining the small size, high on-target activity, and low off-target activity of TadA deaminases. Existing TadCBEs, however, exhibit residual A•T-to-G•C editing at certain positions and lower editing efficiencies at some sequence contexts and with non-SpCas9 targeting domains. To address these limitations, we use phage-assisted evolution to evolve CBE6s from a TadA-mediated dual cytosine and adenine base editor, discovering mutations at N46 and Y73 in TadA that prevent A•T-to-G•C editing and improve C•G-to-T•A editing with expanded sequence-context compatibility, respectively. In *E. coli*, CBE6 variants offer high C•G-to-T•A editing and no detected A•T-to-G•C editing in any sequence context. In human cells, CBE6 variants exhibit broad Cas domain compatibility and retain low off-target editing despite exceeding BE4max and previous TadCBEs in on-target editing efficiency. Finally, we show that the high selectivity of CBE6 variants is well-suited for therapeutically relevant stop codon installation without creating unwanted missense mutations from residual A•T-to-G•C editing.

Base editors are programmable precision genome editing tools that consist of a base-modification enzyme such as a deaminase fused to a programmable DNA-binding domain such as a CRISPR-Cas9 nickase[1,2], a TALE repeat array[3,4],or a zinc-finger array[5,6]. Cytosine base editors (CBEs)[1] enable C•G-to-T•A editing, while adenine base editors (ABEs)[2] enable A•T-to-G•C editing. In contrast with nucleases that generate uncontrolled mixtures of indels, base editors create specified changes at target DNA sequences and do not require double-strand DNA breaks or donor DNA templates[1,2,8]. CRISPR base editors unwind double-stranded DNA (dsDNA), allowing a single strand-specific deaminase to access the DNA strand not paired with the guide RNA, resulting in deamination of C or A

nucleobases within the editing window. Nicking the non-editing DNA strand stimulates its replacement by cellular DNA repair processes to yield a permanently edited base pair[7,8].

Recent efforts have improved the activities[9-11], sequence context compatibilities[9], control over editing window sizes[12], protospacer-adjacent motif (PAM) compatibilities[9,12-14], and size[15] of base editors. Base editing has been used in vivo and ex vivo in animal models to rescue genetic diseases including Hutchinson-Gilford progeria syndrome[16], sickle cell disease[17], spinal muscular atrophy[18], T-cell acute lymphoblastic leukemia[19,20], and others[21,22]. Recently, base editing strategies have entered clinical trials as therapeutics[20,23], with the first positive clinical outcomes[20].

[1]Merkin Institute of Transformative Technologies in Healthcare, Broad Institute of MIT and Harvard, Cambridge, MA, USA. [2]Department of Chemistry and Chemical Biology, Harvard University, Cambridge, MA, USA. [3]Howard Hughes Medical Institute, Harvard University, Cambridge, MA, USA. ✉e-mail: drliu@fas.harvard.edu

The laboratory-evolved deaminase[2,10,14] used in ABEs, TadA*, offers favorable properties for precision genome editing including high on-target activity[14], low off-target editing[10,14,24–26], and small size (166 amino acids) that allows it to be packaged into a single adeno-associated virus (AAV) system[27]. The naturally occurring cytidine deaminases used in CBEs, in contrast, are larger (227 amino acids for the commonly used rAPOBEC1[1]) and suffer from higher Cas-independent DNA and RNA off-target activity and lower on-target editing efficiency[26]. To date, no CBE has been shown to match the most active adenine base editors such as ABE8e in peak editing activity. We hypothesize that this lower editing efficiency of CBEs can be attributed to either lower intrinsic deamination activity or the effects of base excision repair following uracil excision by endogenous uracil glycosylase (UNG). To address these limitations, we and others recently described the first TadA-derived CBEs, which were developed through directed evolution (TadCBEs[28], CBE-Ts[29]) or rational protein engineering (Td-CBEs[30]). These TadA-derived CBEs exhibit low off-target editing and are ~60 amino acids smaller than BE4max, a canonical CBE that uses a natural cytidine deaminase. While some TadA-derived CBEs such as TadCBEs and CBE-Ts have comparable activity to APOBEC1-derived CBEs such as BE4max and evoAPOBEC-BE4max, they retain residual A•T-to-G•C editing at certain positions in the base editing window. Since A•T-to-G•C edits can remove stop codons[21], residual activity limits the utility of TadCBEs for therapeutic stop codon installation. While Td-CBEs offer relatively high product purity, they have substantially lower activity than BE4max and evoAPOBEC-BE4max (see below).

Here, we overcome the limitations of current TadA-derived CBEs through phage-assisted evolution of TadDE, a dual editor that performs both A•T-to-G•C and C•G-to-T•A editing[28], into highly selective TadCBEs. The resulting evolved CBE6 variants show virtually no A•T-to-G•C editing and demonstrate superior C•G-to-T•A editing in mammalian cells when compared side-by-side with all three families of previously reported TadA-derived CBEs. CBE6 editors evolved new mutations in the substrate pocket that directly interact with the target base, as well as mutations at the dimerization interface of TadA. The editors enable highly efficient and cytosine-selective on-target editing with minimal sequence context bias and low off-target editing. Due to their enhanced selectivity and high activity, CBE6 base editors represent state-of-the-art cytosine base editors and are especially advantageous for applications that install stop codons to reduce the levels of proteins associated with increased disease risk (Fig. 1a).

## Results

### Phage-assisted evolution of TadCBEs with improved selectivity

We previously reported the phage-assisted evolution of the cytidine deaminase TadA-CD from TadA-8e, a highly active laboratory-evolved deoxyadenosine deaminase[14,28]. We hypothesized that the selectivity and activity of TadA-CD might be further improved by using an alternative evolutionary starting point, which could enable access to mutations that are inaccessible to highly evolved TadCBEs due to epistasis[31].

We initiated an evolution campaign on TadA-Dual, a dual cytidine and adenosine deaminase used in the dual cytosine and adenine base editor TadDE[28], with the goal of improving the selectivity of this deaminase to exclusively perform cytidine deamination (Fig. 1b). We used phage-assisted continuous evolution (PACE), which maps the stages of traditional directed evolution to the lifecycle of bacteriophage M13 propagating on a culture of *E. coli* host cells[32]. In phage-assisted evolution, the fitness of a gene variant is linked through a genetic circuit to the expression of M13 gIII, which encodes a protein essential to phage propagation. In our circuit, we coupled cytidine deamination activity to gIII expression by fusing T7 RNA polymerase (RNAP) to a bacterial degron[9]. C•G-to-T•A editing activity installs a stop codon in the linker between T7 RNAP and its degron, yielding active T7 RNAP that

transcribes gIII. The mutagenesis plasmid (MP) introduces mutations in the deaminase, and beneficial mutations facilitate phage propagation in the lagoon (fixed-volume vessel), while the less-fit phage are washed out. Phage-assisted non-continuous evolution (PANCE) is an analogous method that relies on manual, discrete dilution of the phage in the lagoons instead of continuous dilution, thus offering a higher likelihood of allowing even modestly beneficial mutations to propagate at the expense of lower evolutionary speed compared to PACE. Stringency in both methods was tuned by altering the lagoon dilution rate and promoter strength upstream of T7 RNA polymerase.

We used a selection circuit we previously developed that penalizes residual adenine base editing[9,28]. In this selection, A•T-to-G•C editing disrupts stop codon installation in the linker between T7 RNAP and its degron, leading to T7 RNAP degradation and no phage propagation[9,28]. This selection thus directs selection pressure to minimize deoxyadenosine deamination activity[28], allowing simultaneous evolution of TadDE for increased CBE activity and reduced ABE activity (Fig. 1c)[28]. Over six passages of PANCE, phage titers continued to increase despite increasing the selection stringency through higher dilution factors and the use of a weaker promoter upstream of T7 RNAP, suggesting that the phage evolved more active or more selective deaminase variants. This first PANCE campaign, in which the phage population underwent a ~$10^{16}$-fold total dilution, yielded converged mutations at the N46 (N46I, N46T) and Y73 (Y73P) positions across different lagoons (Supplementary Figs. 1a and 2). During the course of this work, we were further encouraged by an independent study that reported the importance of the N46 position for target base selectivity[30].

Mutations during PACE arise predominantly from the MP, which promotes all types of substitutions but is biased towards transition mutations[33]. To thoroughly access amino acids that may be less likely to arise through MP-mediated mutagenesis, we constructed a phage library encoding all possible amino acids at TadA position N46 and subjected these variants to a second high-stringency PANCE that used weaker promoters for T7 RNAP (Supplementary Fig. 1b). Persisting phage survived a ~$10^{12}$-fold overall dilution. To further increase stringency, we performed PACE for 118 hours to subject the variants to greater selection pressure from continuous dilution (Fig. 1d, Supplementary Fig. 1c, d). Interestingly, N46L[30], N46V, and N46C eventually converged to N46C at 118 hours, corresponding to an average ~$10^{33}$ fold-dilution. Overall, the final variants that emerged from all evolution campaigns survived an overall dilution of ~$10^{61}$-fold.

Based on the cryogenic electron microscopy (cryo-EM) structure of ABE8e (Protein Data Bank (PDB): 6VPC)[34], we hypothesize that the N46 position determines base selectivity by interacting with the base in the active site to potentially make it more accessible for editing. Mutations at Y73, which is at the dimerization interface of TadA, could impact enzyme assembly, activity, or stability (Fig. 1e). To predict the impact of mutations at position 73 on TadA*, we estimated the energy difference between the TadA* structure with and without Y73P using Rosetta[35]. We found that Y73P stabilizes the TadA* monomer by 11.8 REU (Rosetta Energy Units) compared to Y73S (the original mutation in TadDE), and thus Y73P may enhance the activity of the new TadCBEs (Supplementary Fig. 3).

### Evaluation of activity and selectivity of evolved CBEs in E. coli

To assess the performance of these new deaminases, we first characterized the corresponding TadCBEs in *E. coli*. We developed an *E. coli* plasmid profiling library to interrogate the sequence-context preferences of base editors (Fig. 2a). The new CBEs were tested on a 32-member plasmid library that includes all possible sequence contexts immediately 5' and 3' of a target sequence at protospacer position 6 within the editing window (counting the NGG PAM as positions 21-23). The library was constructed with sequences comprising all nucleotide combinations before and after the target nucleotide, resulting in 16

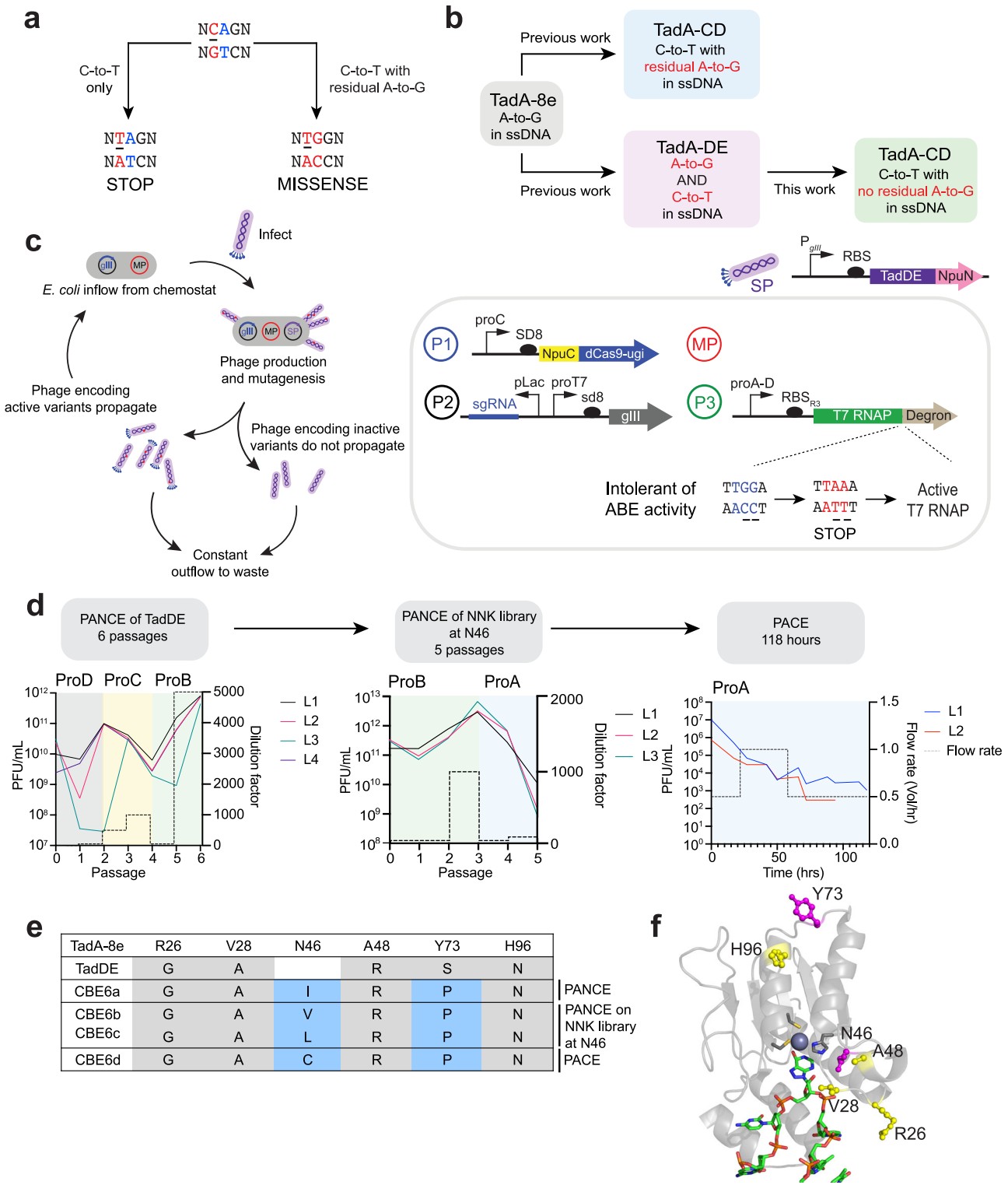

sequences with a target cytosine and 16 sequences with a target adenine. When expressed using the strong ribosome binding site (RBS) SD8[36], the new CBEs showed very high average C•G-to-T•A editing levels of 88% (TadDE N46I Y73P, hereafter designated CBE6a), 95% (TadDE N46V Y73P, hereafter designated CBE6b), 95% (TadDE N46L Y73P, hereafter designated CBE6c), and 88% (TadDE N46C Y73P, hereafter designated CBE6d), which are comparable or superior to editing by TadCBEd (88%), a previous state-of-the-art CBE[28] (Fig. 2b, Supplementary Fig. 4). When expressed using the weaker ribosome binding site sd5[36], the new TadCBEs showed average C•G-to-T•A

editing levels of 82% (CBE6a), 90% (CBE6b), 95% (CBE6c), and 89% (CBE6d), again outperforming TadCBEd (78%) (Supplementary Fig. 5). Next, we assessed the sequence-context preference of the cytosine base editors using the SD8 RBS. While CBE6a and CBE6d showed similar sequence-context preferences as TadCBEd, disfavoring 5′ AC and 5′ GC, two variants (CBE6b and CBE6c) performed C•G-to-T•A editing equally well (over 80% editing) at every possible sequence context (Fig. 2b, Supplementary Fig. 4).

To assess cytosine versus adenine deamination selectivity, we analyzed residual A•T-to-G•C editing in the library. When expressed in

**Fig. 1 | Development of a highly active and selective cytosine base editor from a TadA dual base editor using phage-assisted evolution. a** An active and selective cytosine base editor, but not one with residual adenine base editing activity, can cleanly install stop codons into target genes. **b** Schematic of the evolution of a cytosine base editor from a TadA-derived dual base editor (TadDE)[29]. **c** Diagram depicting phage-assisted continuous evolution (PACE, left) and the selection circuit used in this study (right). A continuous flow of *E. coli* host cells with the selection circuit and a mutagenesis plasmid (red) are infected by selection phage encoding a deaminase (SP). In the selection circuit, phage propagation is linked with gIII expression (P2), which can only be transcribed with active T7 RNA polymerase. T7 RNA polymerase (P3) is fused to a C-terminal degron, and the deaminase must perform C•G-to-T•A editing to install a stop codon before the degron to generate active T7 RNA polymerase. In the event of phage infection, the full base editor is reconstituted using a split-intein system (P1), and mutations accumulate in the deaminase. Beneficial mutations lead to phage propagation and enrichment in the lagoon, while the less-fit phage are unable to propagate and are washed out by the constant outflow. **d** Evolutionary trajectory of an active and selective cytosine base editor from TadDE. Phage-assisted non-continuous evolution (PANCE) was performed on TadA-DE until phage titers increased despite higher stringency. The resulting genotypes identified a conserved mutation at position N46 in TadA, so an NNK library was constructed to diversify this position, and PANCE was performed on the resulting variants. PACE was performed for >100 hrs on the resulting variants from both PANCE experiments. Dilution factors are indicated on the right y-axis. Relative promoter units (normalized to proD) for proA, proB, proC, and proD are as follows: 0.030, 0.119, 0.278, and 1.000, respectively[55]. **e** Mutation table from evolved deaminases showing conserved mutations. **f** Cryo-EM structure of ABE8e (PDB: 6VPC) with mutations labeled. New mutations are highlighted in magenta, and mutations inherited from TadDE are highlighted in yellow. Source data are provided as a Source Data file.

*E. coli* using the strong SD8 RBS, TadCBEd demonstrated residual A•T-to-G•C editing (average of 8%) at protospacer position 6, especially for 5′ C and 5′ T sequence contexts, consistent with our previous report[28]. In contrast, we identify several CBE6 base editors that show residual A•T-to-G•C editing below the high-throughput sequencing limit of detection (average of < 0.1%), regardless of the sequence context. Thus, these new TadDE-evolved CBEs offer substantially higher product purities than TadCBEd. At position 6 in the protospacer using SD8, the ratio of the new CBEs for C•G-to-T•A editing over A•T-to-G•C editing exceeded 990-fold in all cases, compared to 10.6-fold for TadCBEd, an improvement of at least ~100-fold.

To characterize the base editing window of the new CBEs, all four CBE6s were tested in *E. coli* on a 448-member target site library that includes all possible 5′ and 3′ sequence contexts of a target C or A ranging from positions 1–14 of the protospacer. To maximize observed differences in activity, a weaker RBS (sd2) was used. The new TadCBEs exhibited an editing window−defined as the range where the average editing is at least 20% of the average peak editing−centered near protospacer position 6 and ranging from positions 4-8 (Fig. 2c, Supplementary Figs. 8 and 9), slightly larger than the editing window of TadCBEd, which ranges from positions 5-7. As editing window size and activity are often correlated, we recommend the CBE6 variants especially for applications that need high editing levels and do not require an especially narrow editing window[12]. Averaged across positions 4-8 of the editing window, the selectivity ratio of the new TadCBEs for C•G-to-T•A editing over A•T-to-G•C editing ranged from 27- to 86-fold, compared to 16-fold for TadCBEd.

## Reversion analysis

To determine the contribution of each mutation to achieving CBE selectivity and activity, reversion analysis was performed in which each mutation was added to the starting point (TadDE) in a successive fashion, and each resulting variant was characterized in *E. coli*. We found that adding N46I, N46V, N46L, or N46C were all sufficient to remove A•T-to-G•C editing from TadDE, decreasing the A•T-to-G•C editing efficiency from 75% to an average below 0.1%. Furthermore, the addition of Y73P was necessary to increase C•G-to-T•A editing levels further, leading to an additional 20-53% of sequencing reads edited for the most difficult sequence context tested (Supplementary Fig. 10).

Next, we added the N46 and Y73P mutations to TadCBEd, which was evolved from ABE8e using the same selection circuit[28]. The addition of N46I removed the residual A•T-to-G•C editing from TadCBEd, but the N46I and Y73P mutations were detrimental to C•G-to-T•A editing, decreasing average editing efficiencies by 1.3-fold when only N46I is added and 1.3-fold when both mutations are added (Supplementary Fig. 11). These data indicate that the evolved mutations in TadDE−but not those in TadCBEd−provide the genetic context that supports the beneficial effects of N46I and Y73P. Thus, these new deaminase variants likely did not emerge during PACE experiments that gave rise to TadCBEd because the N46I and Y73P mutations have an epistatic relationship with mutations in TadCBEd.

## Evaluation of activity and selectivity in mammalian cells

Next, we tested the CBE6 variants at a variety of endogenous genomic sites in human cells and compared side-by-side their performance with that of TadCBEd[28], Td-CBEmax[30], and CBE-T1.52[29], the best-performing TadA-derived CBEs recently described by three groups. The new deaminases were fused to SpCas9 or eNme2-C Cas9 nickase domains in the BE4max architecture[11] and transfected into HEK293T cells, along with a plasmid encoding an sgRNA. Using SpCas9, the new CBEs showed similar or superior average peak editing frequencies of 55–59% compared to TadCBEd (average peak editing of 54%; *P* value compared to CBE6a > 0.05, *P* value compared to CBE6b > 0.05), Td-CBEmax (25%; *P* value compared to CBE6a < 0.0001, *P* value compared to CBE6b < 0.0001), and CBE-T1.52 (44%; *P* value compared to CBE6a < 0.05, *P* value compared to CBE6b < 0.05) (Fig. 3a, Supplementary Fig. 12, Supplementary Fig. 23). Note that we selected CBE-T1.52 for comparison because of its high activity and selectivity among the CBE-T variants. The CBE6 variants also displayed superior selectivity for cytosine over adenine at the SpCas9 sites. While the new CBEs showed residual A•T-to-G•C peak editing efficiencies of <0.1-0.1% (CBE6a), <0.1-0.6% (CBE6b), <0.1-0.3% (CBE6c), and <0.1%-1.2% (CBE6d) at all SpCas9 sites that were screened, the previously described TadA-derived CBEs TadCBEd, CBE-T1.52, and Td-CBEmax showed peak A•T-to-G•C editing efficiencies ranging from 4.7-67% (*P* value compared to CBE6a < 0.01, *P* value compared to CBE6b < 0.01), 0.6-11% (*P* value compared to CBE6a < 0.05, *P* value compared to CBE6b < 0.05), and 0.2-12% (*P* value compared to CBE6a < 0.05, *P* value compared to CBE6b < 0.05), respectively (Supplementary Fig. 23). We speculate that differences in the observed residual A•T-to-G•C editing in mammalian cells compared to the residual A•T-to-G•C editing *E. coli* as reported above could be due to differences in their deoxyinosine repair pathways[37].

We constructed CBE6 variants using the eNme2-C Cas9 nickase[38] to assess compatibility with an alternative Cas9 domain (PAM = $N_4CN$). The use of eNme2-C Cas9 also impedes the activity of the deaminase domain compared to SpCas9 as we previously showed with TadCBEs[28]. Across four target sites in HEK293T cells, the new CBEs using eNme2-C Cas9 offered superior average peak C•G-to-T•A editing efficiencies of 28-38%, an improvement over TadCBEd (average peak editing of 25%; *P* value compared to CBE6a > 0.05, *P* value compared to CBE6b > 0.05) (Fig. 3b, Supplementary Fig. 13). Encouragingly, these average editing efficiencies are comparable to or higher than that of ABE8e with eNme2-C Cas9 (29%) and exceed by approximately 6-fold the observed average editing efficiency of Td-CBEmax (5%; *P* value compared to CBE6a < 0.0001, *P* value compared to CBE6b < 0.0001) and CBE-T1.52 (6%; *P* value compared to CBE6a < 0.0001, *P* value compared to

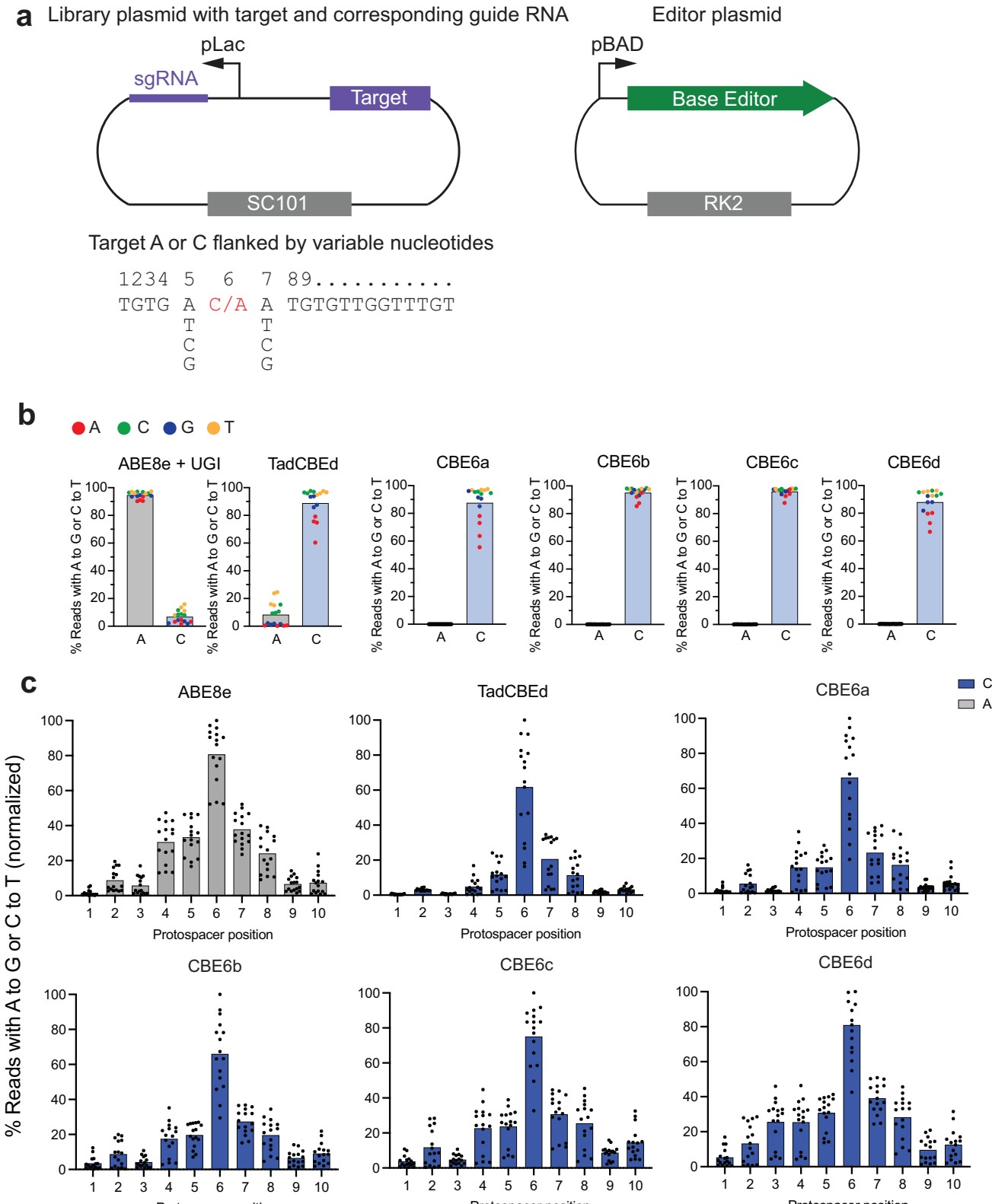

**a** Library plasmid with target and corresponding guide RNA

Editor plasmid

Target A or C flanked by variable nucleotides

**b** ● A ● C ● G ● T

**c**

CBE6b < 0.0001) using eNme2-C Cas9 domains at the same sites (Fig. 3b).

These findings collectively establish that the new CBE6s offer comparable or higher activity than BE4max, evoAPOBEC-BE4max, and the three previously reported TadA-derived CBEs, but with virtually no detected A•T-to-G•C editing. The benefits of the CBE6 variants are especially pronounced when using a non-SpCas9 targeting domain.

**Characterization of Cas-independent and Cas-dependent off-target activity of new CBEs**

Highly active gene editing agents are especially prone to off-target editing, resulting in undesired mutations in genomic DNA or in RNA[26,39]. Off-target base editing can occur through Cas-dependent mechanisms, in which Cas9 engages non-target DNA sequences similar to the target sequence, or through Cas-independent mechanisms, in

**Fig. 2 | Profiling the activity and sequence context specificity of CBE6 variants in *E. coli*. a** Schematic of a 32-member library that varies the 5′ and 3′ sequence contexts of a target edit at position 6 within the editing window of an *E. coli* protospacer. **b** Bar values indicate the average activity of CBE variants when tested on a library of 32 substrates designed to contain the target base (A or C) at protospacer position 6 with all possible combinations of flanking nucleotides. Dots represent the average percentage of sequencing reads containing the specified edit (A•T-to-G•C or C•G-to-T•A) for each of the 16 sequence contexts (each dot represents an average of *n* = 3 independent biological replicates). The dots are colored according to the 5′ upstream base (A, red; C, green; G, blue; T, yellow). **c** Bar values indicate the average C•G-to-T•A editing efficiency of CBE variants or A•T-to-G•C activity of ABE8e when tested on a library of 448 substrates designed to contain the target base (A or C) at protospacer positions 1-10 with the 5′ and 3′ base varied as A, T, C, or G, normalized to the highest C•G-to-T•A activity for CBE variants or A•T-to-G•C activity for ABE8e. Dots represent the average percentage of sequencing reads containing the specified edit (A•T-to-G•C or C•G-to-T•A) for each of the 224 sequence contexts (each dot represents an average of *n* = 2 independent biological replicates). Full data for C•G-to-T•A and A•T-to-G•C activity are provided in Supplementary Figs. 8 and 9. Source data are provided as a Source Data file.

which the deaminase domain operates on other transiently single-stranded DNA sequences independent of Cas protein engagement[26,40].

We performed Cas-independent DNA and RNA off-target analyses on the new CBE6s both with and without V106W. We previously showed that the addition of V106W to TadA variants reduces DNA and RNA off-target activity of ABEs with little or no decrease in on-target editing efficiency[14,41]. V106W was reported as a mutation that reduces off-target RNA deamination by weakening deaminase binding to RNA through steric occlusion[41]. V106W also decreases off-target editing of DNA, perhaps by a similar mechanism (Supplementary Figs. 16–22). However, V106W largely preserved on-target DNA editing activity, possibly due to the high effective concentration of the target DNA substrate that is enforced by fusion to Cas9.

Using the orthogonal R-loop Cas-independent DNA off-target assay[26], we observed that the new CBE6s have similar low levels of DNA off-target activity (average 0.2-0.7%) as TadCBEd (0.5%; *P* value compared to CBE6a > 0.05, *P* value compared to CBE6b > 0.05), which are lower than that of BE4max (average of 1.1%; *P* value compared to CBE6a < 0.05, *P* value compared to CBE6b > 0.05) and evoAPOBEC (average of 1.0%; *P* value compared to CBE6a < 0.01, *P* value compared to CBE6b > 0.05) (Fig. 4a, Supplementary Fig. 26). With the addition of V106W, on-target editing levels are only slightly decreased (1.3-fold decrease for CBE6a; 1.1-fold decrease for CBE6b; 1.05-fold decrease for CBE6c; 1.01-fold average decrease for CBE6d), but Cas-independent DNA off-target editing levels are greatly decreased (all to ≤0.1%) (Fig. 4a). *P* values were <0.01 and <0.01 for comparing CBE6a V106W to BE4max and evoAPOBEC, respectively and <0.05 and <0.001 for comparing CBE6b V106W to BE4max and evoAPOBEC, respectively. (Supplementary Fig. 26).

TadCBEd offers lower Cas-independent RNA off-target editing than BE4max and evoAPOBEC[28]. Here, off-target RNA editing analysis revealed that the new CBE6s edited an average of 0.1% of cytosines across three transcripts prone to off-target ABE editing (*CTNNB1*, *IP9O*, and *RSL1D1*), comparable to the average off-target RNA editing of 0.1% for TadCBEd (*P* value compared to CBE6a < 0.05, *P* value compared to CBE6b > 0.05) (Fig. 4b). The addition of V106W to the new CBEs slightly decreased the average RNA off-target editing of cytosines to <0.1%. Additionally, the new CBE6 variants showed <0.1% A•T-to-G•C editing across transcripts, below the limit of detection of HTS.

To characterize Cas-dependent off-target editing of the new CBE6s, we investigated 22 previously documented off-target sites for SpCas9 base editors and sgRNAs targeting *HEK3*, *HEK4*, *EMX1*, and *BCL11A* (Supplementary Figs. 16–19). In general, the new CBE6s showed comparably low levels of Cas-dependent off-target editing as those of TadCBEd across the 22 off-target sites (Supplementary Figs. 27–30). Cas-dependent off-target editing is more easily addressed than Cas-independent off-target editing since the former can be ameliorated by varying guide RNA sequence or length, the PAM sequence targeted, and the Cas domain. As Cas-dependent off-target editing can limit the therapeutic utility of CRISPR gene editing agents, high-fidelity Cas proteins that are known to engage fewer off-target loci may improve therapeutic relevance by reducing Cas-dependent off-target editing[38].

Translocations or other chromosomal abnormalities can occur if a DNA single-strand break is converted to a double-strand break during cell replication[42]. Previous work has shown that potential translocations generated by base editors are correlated with the fraction of indels detected[42]. CBE6 indel levels are low and comparable to previously reported CBEs and ABEs, suggesting that the CBE6 variants will not generate more translocations than previously reported base editors (Supplementary Figs. 14 and 15). Translocations were virtually undetected via ddPCR in a prior study using base editing with optimized reagents[43].

## Stop codon installation at therapeutically relevant genomic sites in human cells

To demonstrate the utility and performance of these new CBEs, we used them to install stop codons at several therapeutically relevant sites in the genome. Gene knockout and gene silencing are strategies being applied in clinical trials to suppress the levels of proteins associated with disease or inactivate gain-of-function mutant genes[21]. Installation of a premature stop codon by cytosine base editing can achieve these goals while avoiding the complex mixtures of uncontrolled indel products that result from nuclease-mediated gene knock out[44,45]. The lack of residual A•T-to-G•C editing is important for this application because A•T-to-G•C editing of either strand of an installed TAG, TAA, or TGA nonsense codon would convert it to a missense mutation, restoring undesired readthrough of a mutated target.

To test the ability of the new CBEs to perform clean premature stop codon installation at therapeutically relevant genomic loci, we identified several genomic sites within *PCSK9* that were previously studied to lower LDL cholesterol levels[21]. We designed three sgRNAs that install stop codons at positions in *PCSK9* that generate protein-truncating variants with potential therapeutic utility[21]. We electroporated synthetic guide RNA and mRNA encoding CBEs into patient-derived fibroblasts. We then measured cytosine and residual adenine base editing activity at these *PCSK9* target sites in patient-derived fibroblasts, comparing the new CBEs to previously reported TadCBEs, BE4max, evoFERNY, and YE1. Across the three target sites for stop codon installation, the new CBE6s resulted in virtually no detected (average of 0.1%) residual A•T-to-G•C editing and also generally yielded the highest editing levels, averaging 41-53% at the target C (Fig. 5). TadCBEd yielded an average editing level of 27% (*P* value compared to CBE6a > 0.05, *P* value compared to CBE6b < 0.001) at the target C with 4% residual A•T-to-G•C editing (*P* value compared to CBE6a < 0.0001, *P* value compared to CBE6b < 0.0001), which converts the installed TAG stop codon to a TGG missense codon (Supplementary Fig. 25). As a result, an average of 16% of the stop codons installed by TadCBEd were converted to missense codons. While CBE-T1.52 displayed higher selectivity than TadCBEd, it showed lower editing efficiency (34%; *P* value compared to CBE6a > 0.05, *P* value compared to CBE6b < 0.05) at the target C than the CBE6 variants and still caused an average of 0.9% residual A•T-to-G•C editing at these sites (*P* value compared to CBE6a < 0.01, *P* value compared to CBE6b < 0.01) (Supplementary Fig. 25). BE4max averaged 32% on-target editing (*P* value compared to CBE6a > 0.05, *P* value compared to CBE6b < 0.01) but induces much

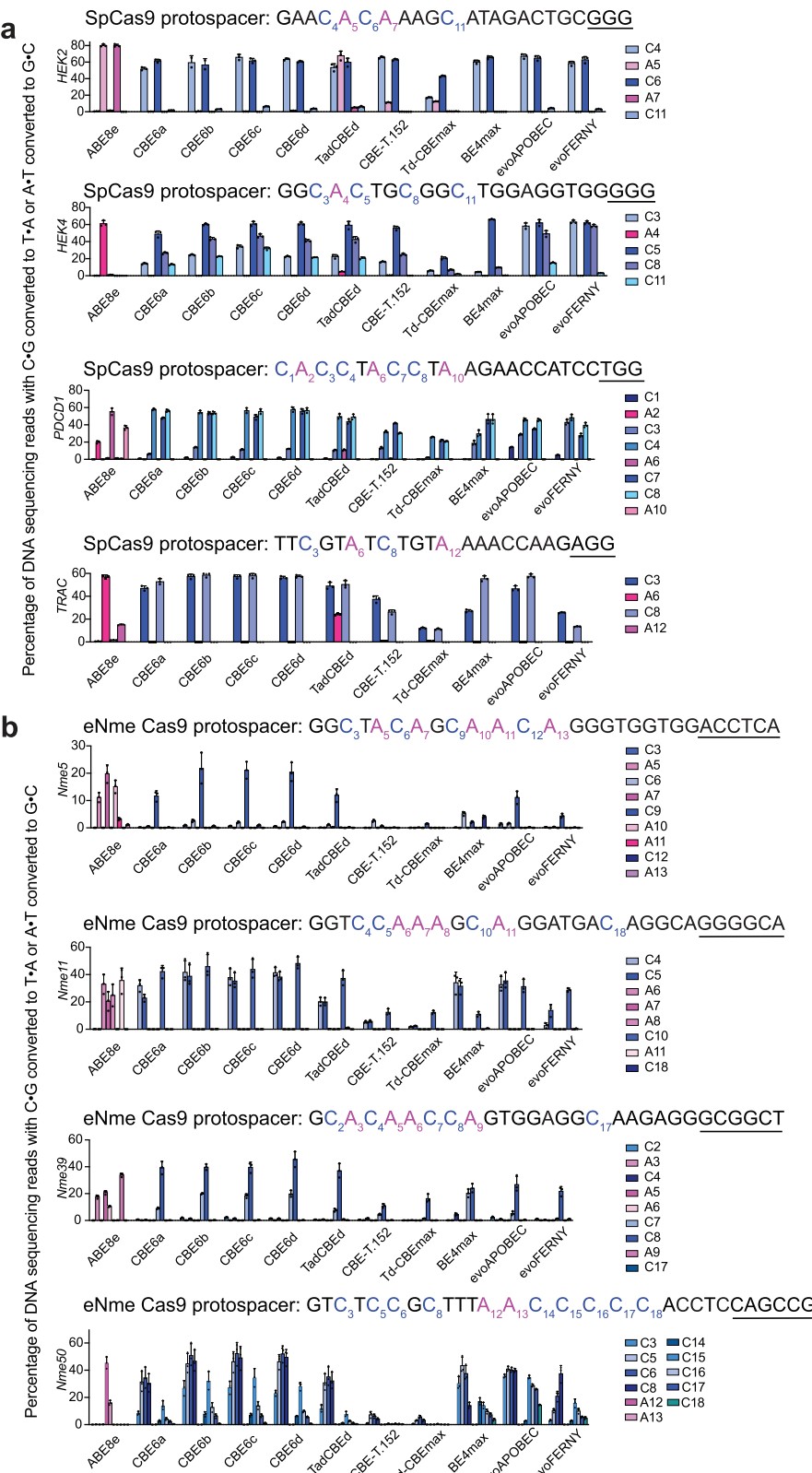

**Fig. 3 | Comparison of CBE6 variants with existing CBEs in mammalian cells.**
**a** CBE6 variants or existing cytosine base editors, all using SpCas9 nickase domains in the BE4max architecture, were transfected into HEK293T cells with guide RNAs targeting three protospacers. Data are presented as mean values ± SD. Dots represent individual values from *n* = 3 independent biological replicates. PAM sequences are underlined. HEK293T site 2 is abbreviated *HEK2*, and HEK293T site 4 is abbreviated *HEK4*. **b** CBE6 variants along with existing cytosine base editors using

eNme2-C Cas9 nickases in the BE4max architecture were transfected into HEK293T cells with guide RNAs targeting three protospacers. Data are presented as mean values ± SD. Dots represent individual values from n = 3 independent biological replicates. PAM sequences are underlined. Full data for C•G-to-T•A and A•T-to-G•C activity are in Supplementary Figs. 12 and 13. Source data are provided as a Source Data file.

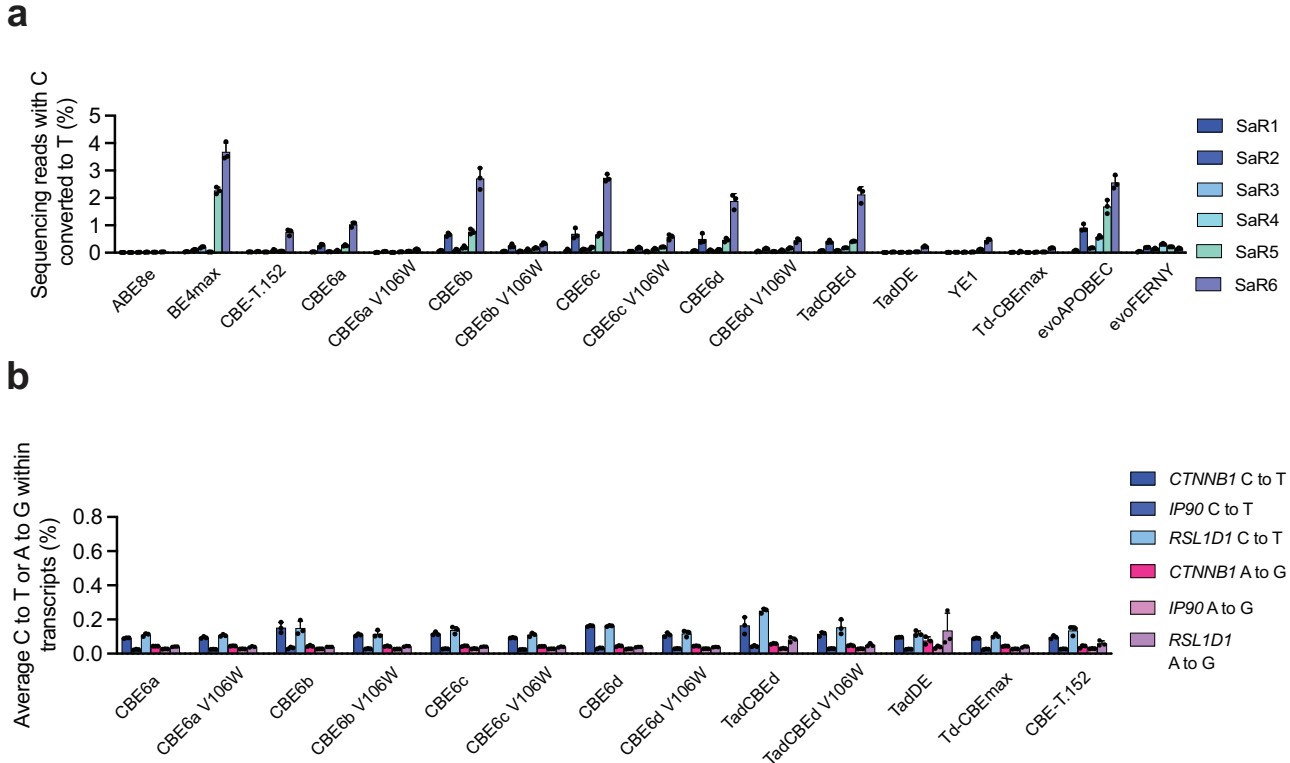

**Fig. 4 | Cas9-independent DNA and RNA off-target editing by CBE6 variants.**
**a** Average Cas9-independent off-target editing across all cytosines for six orthogonal R-loops (SaR1–SaR6) generated by a dead *S. aureus* Cas9. Data are presented as mean values ± SD. Dots represent individual values from independent biological replicates. **b** Off-target RNA editing. Base editors were transfected into HEK293T cells and harvested after 48 hours. cDNA was synthesized, and high-throughput sequencing was used to analyze *CTNNB1*, *IP90*, and *RSL1D1*. Data are presented as mean values ± SD. Dots represent individual values from independent biological replicates. Source data are provided as a Source Data file.

higher off-target activity, as shown previously[26,28] (Supplementary Fig. 25).

Taken together, these findings indicate that the new CBE6 base editors offer enhanced activity, C versus A deamination selectivity, and target specificity compared with previously reported cytosine base editors. When residual A•T-to-G•C editing must be kept to an absolute minimum, we recommend CBE6a (TadDE N46I Y73P), though we note that CBE6a retains sequence context preferences that disfavor 5' AC and 5' GC sequences. We recommend CBE6b (TadDE N46V Y73P) for general cytosine base editing applications, especially when Cas domains other than SpCas9 are used. For cytosine base editing applications in which off-target editing must be strictly minimized, we recommend using CBE6a-V106W and CBE6b-V106W.

## Discussion

Here we report the evolution and characterization of new CBE6 variants with high C•G-to-T•A editing activity and virtually no residual A•T-to-G•C activity. These variants did not emerge from the evolution of previously reported TadCBEs, but instead were evolved from the dual adenine and cytosine base editor TadDE, suggesting the value of this starting point for CBE evolution trajectories. The new CBE6 variants are >100-fold more selective for C•G-to-T•A editing than TadCBEd when tested at protospacer position 6 in *E. coli*. We show that the residue at position 46 near the target base confers selectivity for cytidine deamination, and position 73 at the dimerization interface aids in increasing editing efficiency. In both *E. coli* and mammalian cells, the new CBE6 variants show virtually no residual A•T-to-G•C editing and outperform current CBE variants in on-target editing efficiency. Cas9-independent DNA and RNA off-target editing levels, as well as Cas9-dependent off-target editing levels, are similar to those of

TadCBEd and lower than that of BE4max and can be further reduced without substantially lowering on-target editing efficiencies by adding the V106W mutation.

The new CBE6 variants offer substantial benefits when installing stop codons at genomic sites for gene knockout by avoiding the undesired creation of missense codon byproducts, as demonstrated by editing *PCSK9* in patient-derived fibroblasts. In addition to enhancing precision gene editing applications, the high editing efficiencies and very high selectivities of CBE6 variants may also benefit genetic screens that use base editors to create libraries of many gene variants to uncover structure-function insights[46–49].

## Methods

### Molecular cloning

All plasmid construction was completed through Gibson assembly or SapI-Golden Gate methods (New England Biolabs). PCR amplification was performed using Phusion U Green Hot Start II DNA polymerase (Thermo Fisher Scientific) and nuclease-free water (Qiagen). Cloning products were transformed into Mach1 chemically competent *E. coli* cells (Thermo Fisher Scientific). Selection antibiotics were employed at the indicated final concentrations: carbenicillin at 100 μg/ml, spectinomycin at 50 μg/ml, kanamycin at 50 μg/ml, chloramphenicol at 25 μg/ml, and tetracycline at 10 μg/ml.

For Sanger sequencing (Quintara Biosciences), plasmid DNA was amplified using the Illustra Templiphi 100 Amplification Kit (GE Healthcare Life Sciences). Sequence-validated plasmids intended for bacterial transformation were purified using the Spin 2.0 Miniprep Kit (Qiagen), and the Plasmid Plus Midi Kit (Qiagen) were used for plasmids for bacterial transformation and mammalian transfection, respectively. The concentrations of plasmids were determined using

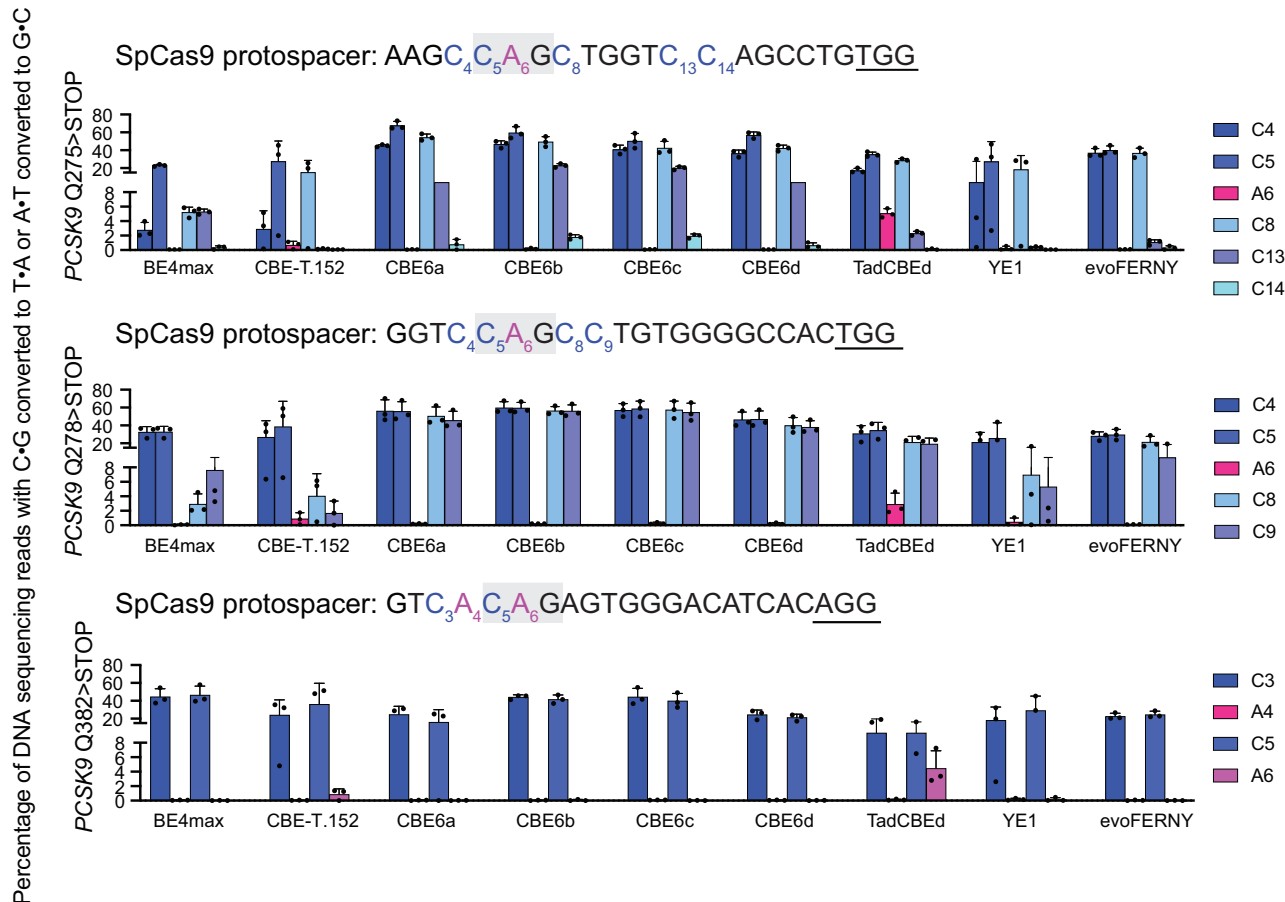

**Fig. 5 | Stop codon installation at therapeutically-relevant loci by CBE6 variants in fibroblasts.** CBE6 variants were used to install stop codons in *PCSK9*, a therapeutic strategy for lowering LDL cholesterol levels. The gray boxes indicate the desired location of stop codon installation. Data are presented as mean values ± SD. Dots represent individual values from independent biological replicates. PAM sequences are underlined. Source data are provided as a Source Data file.

NanoDrop technology. Plasmids encoding CBE6 variants are available from Addgene.

## Bacteriophage cloning

To perform Gibson assembly of the phage, PCR fragments (1 uL each) were combined in a final volume of 4 μl. Following Gibson assembly, the reaction mixture was introduced into chemically competent S2208 *E. coli* host cells, defined as S2060 *E. coli* host cells harboring pJC175e[32], with a transformation volume of 100 μl. These cells, capable of activity-independent phage propagation, were cultured for 5 hours at 37 °C with agitation in antibiotic-free 2×YT media then centrifuged at 10,000 *g* for 10 minutes. Clonal phage populations were isolated by performing plaque assays as described below. Individual plaques were then cultivated in DRM media (prepared from United States Biological CS050H-001/CS050H-003) for a duration of 6–8 hours. To eliminate *E. coli* contaminants, the bacterial culture was centrifuged at 6000 g for 10 minutes, and the resulting supernatant was removed for use. For subsequent sequencing, the gene of interest within the phage was amplified using primers AB1793 (5′-TAATGGAAACTTCCTCAT-GAAAAAGTCTTTAG) and AB1396 (5′-ACAGAGAGAATAACATAAAAA-CAGGGAAGC), followed by Sanger sequencing. These primers (Integrated DNA Technologies) anneal to the phage backbone and flank the gene of interest. Finally, phage samples were stored at 4 °C.

## Transformation using chemically-competent cells

For all phage propagation, PANCE, and PACE experiments, strain S2060 was used. Competent cells were prepared by diluting an overnight culture 100-fold into 25 ml of 2×YT media (United States

Biological) with tetracycline and streptomycin. The culture was then grown at 37 °C with gentle shaking at 230 r.p.m. until reaching an $OD_{600}$ of approximately 0.4–0.6 then pelleted by centrifugation at 4000 *g* for 10 minutes at 4 °C. To create competent cells, the resulting cell pellet was resuspended in 2.5 ml of TSS (LB media supplemented with 5% v/v DMSO, 10% w/v PEG 3350, and 20 mM $MgCl_2$), divided into 100-μl aliquots, flash-frozen in liquid nitrogen, and stored at −80 °C.

The transformation process involved the use of 100 μl of competent cells that were thawed on ice and combined with a mixture of plasmids (1 μl each, with a maximum of three plasmids per transformation) in 20 μl of 5× KCM solution (500 mM KCl, 150 mM $CaCl_2$, and 250 mM $MgCl_2$ in water) along with 80 μl of water. The mixture was then incubated on ice for 15 minutes. A heat-shock step was performed at 42 °C for 90 seconds, after which 800 μl of SOC media (New England Biolabs) was added to rescue the cells. The cells were allowed to recover at 37 °C at 230 r.p.m. for 0.5–1.5 hours. Subsequently, the transformed cells were plated on 2×YT media containing 1.5% agar (United States Biological) and appropriate antibiotics to be incubated at 37 °C overnight.

## Plaque assays

In order to facilitate the propagation of phage without relying on their activity, they were subjected to plaque formation on S2208 *E. coli* host cells[32]. A culture of host cells, whether freshly prepared or stored at 4 °C for a maximum of 3 days, underwent a 50-fold dilution in DRM supplemented with suitable antibiotics. Subsequently, the cells were cultivated at 37 °C until reaching an $OD_{600}$ measurement of 0.4–0.8.

To establish different concentrations of phage stocks, serial dilutions were performed using DRM, with each dilution being tenfold more than the previous one. For the creation of plaquing plates, a mixture consisting of molten 2×YT agar (comprising 1.5% agar at 55 °C) and Bluo-gal (Gold Biotechnology), with a final concentration of 0.08% Bluo-gal, was dispensed into the wells of a 24-well plate, with each well receiving 1 ml of the mixture and left undisturbed at room temperature until solidification.

To prepare the top agar, a mixture was made by combining 2×YT medium and molten 2×YT medium agar (at a concentration of 1.5%) in a ratio of 3:2. This mixture was then stored at 55 °C until it was ready for use. For the plaquing process, 100 µl of cells were combined with 10 µl of phage in 2-ml library tubes (VWR International) to which 300 µl of warm top agar was added. After briefly mixing, this was immediately pipetted onto the solid agar medium in one of the wells of the 24-well plate. The top agar was left undisturbed to solidify at 25 °C. The plates were then incubated at 37 °C overnight without being inverted. The quantification of phage titers was accomplished by counting the number of blue plaques and using the following formula: titer (in $\frac{PFU}{ml}$) = (#of plaques in quadrant) (dilution factor of quadrant)(100).

To prepare S2060 cells harboring the AP and CP plasmids of interest, the aforementioned procedure was followed. To determine phage fold enrichment, the S2060 cells harboring the plasmids of interest were inoculated into DRM overnight then diluted 50-fold into fresh DRM and cultivated at 37 °C until reaching an $OD_{600}$ of 0.4–0.8 the following day. These cells were dispensed into the wells of a 96-well plate, with each well containing 1 ml of culture (Axygen). Subsequently, phage with a known titer were added to achieve an input concentration of $10^5$ plaque-forming units per milliliter (PFU ml$^{-1}$). The cultures were incubated overnight at 37 °C with continuous shaking at 230 r.p.m.

Following the incubation period, the plates were centrifuged at 4000 g for 10 minutes to separate the cells from the phage, resulting in the phage being present in the supernatant. The supernatants were then subjected to titering using the plaquing method described earlier. To determine the fold enrichment, the titer of propagated phage in the output was divided by the titer of input phage.

### PANCE
PANCE experiments were conducted following established protocols[50]. Chemically competent S2060 host cells, transformed with AP and CP, were prepared as described above. These competent host cells were then transformed with a mutagenesis plasmid (MP6)[33] and plated on 2×YT agar supplemented with 100 mM glucose and the suitable antibiotics. Strains containing MP6 are grown with glucose-containing media to repress the arabinose promoter and are not recommended to be stored for more than 1 week. Subsequently, three colonies were selected and transferred to individual wells of a 96-well plate containing 1 ml of DRM and suitable antibiotics. The colonies were resuspended and underwent ten-fold serial dilution, repeated eight times in DRM. The plate was sealed with a porous film and incubated at 37 °C with shaking at 230 r.p.m. for 16-18 hours. Wells with dilutions with an $OD_{600}$ of approximately 0.4 were combined with 20 mM arabinose to induce mutagenesis and pipetted into the necessary number of 1 ml lagoons in a 96-well plate. Selection phage at the specified dilution were added to the cultures, which were incubated overnight at 37 °C then harvested by centrifugation at 4000 g for 10 minutes. 150 µl of the resulting supernatant with the evolved phage was transferred to a 96-well PCR plate, sealed with foil, and stored at 4 °C. The phage were utilized for subsequent passages. Phage titers were determined either by qPCR using a previously reported protocol or by plaquing.

### PACE
PACE experiments were conducted in accordance with previously published protocols[50]. Host cells harboring the mutagenesis plasmid

were prepared, then twelve colonies were transferred into individual wells containing 1 ml of DRM and suitable antibiotics of a 96-well plate. The colonies were resuspended and underwent ten-fold serial dilution, repeated eight times in DRM. The plate was sealed with a porous film and incubated at 37 °C with shaking at 230 r.p.m. for 16–18 hours. Wells with dilutions with an $OD_{600}$ of approximately 0.4 were combined then added to a chemostat with 80–100 ml of DRM in a warm room. The chemostat was incubated until reaching an $OD_{600}$ of approximately 0.4–0.8, with continuous dilution using fresh DRM at a rate of 1–1.5 chemostat volumes per hour to maintain constant cell density.

Prior to infection, 15 ml of culture from the chemostat was added to each lagoon, which were pre-induced with 10 mM arabinose for a minimum of 1 hour. 250 mM arabinose was continuously added to the lagoons at a rate of 0.6 mL per hour. Selection phage infection was initiated in the lagoons with an initial titer of $10^7$ PFU ml$^{-1}$. The lagoon dilution rates were gradually increased over time for higher selection pressure. 1 mL samples were collected from the lagoon waste lines at specified time intervals, centrifuged at 6,000 g for 8 minutes, and the resulting supernatant containing the evolved phage was stored at 4 °C. Phage titers were calculated after plaquing with S2208 *E. coli* host cells[32]. PCR amplification using the AB1793/AB1396 primer pair followed by Sanger sequencing were used to confirm the sequences of plaques.

### E. coli profiling assay
To generate the library, a 448-member single-stranded DNA library (IDT oligopools) was designed to contain the target base (A or C) at protospacer positions 1-14 with the 5' and 3' base varied as A, T, C, or G. Each library member contains a unique molecular identifier (UMI) barcode (Supplementary Data 2). The single-stranded oligos were amplified for three cycles with the primer pair MN1591/MN1592 with KAPA polymerase using 1.5 nM template in a reaction volume of 200 µl with an annealing temperature of 68 °C and an extension time of 3 min. The PCR product was purified (Qiagen) and assembled into BamHI/EcoRI-digested plasmid MNp553 using Gibson (NEB). Following purification with Glyco-blue (Thermo Fisher), the library was transformed into NEB 10-beta electrocompetent cells. Dilutions of cells were plated immediately to calculate library size, and then the remaining transformants were grown overnight in carbenicillin to select for transformants. The following day, the library plasmid was purified by Midiprep (Qiagen).

In parallel, electrocompetent NEB10-beta cells containing the indicated editor plasmid of interest were prepared following growth in DRM to suppress expression. 40 µl of electrocompetent cells containing the editor was then electroporated with 100 ng library plasmid, rescued in 1 ml S.O.C. media for 5 min, diluted in 35 ml DRM, and grown overnight with spectinomycin, carbenicillin, and 30 mM arabinose to induce editor expression. After 16 h growth at 37 °C with shaking at 200 rpm, the plasmids were isolated by Midiprep. 1 µl plasmid was used as a template for PCR1 and HTS analysis as indicated below.

To analyze editing results for the library, sequencing reads were demultiplexed using MiSeq Reporter (Illumina) and then sorted into target amplicons using SeqKit. The output was then sequenced using CRISPResso2. The CRISPResso2 output was analyzed using a Python script adapted from Doman et al.[26] and Zhang et al.[51]. The output was then plotted and analyzed in PRISM 10.

To determine selectivity for cytosine over adenine deamination for each editor, we calculated the average cytosine editing efficiency and the average adenine editing efficiency at positions 4–8 in the editing window. We then computed the geometric mean of the ratio of average cytosine editing to average adenine editing at each position.

Sequence logos were generated for each editor to quantify the relative editing efficiency at each sequence context around the edited C or A. To calculate the relative editing for each context, editing

efficiency was summed over edits with a particular base 5′ or 3′ of the edited base for all positions in the window, then normalized by dividing by the total editing over all contexts. This process yielded a frequency value for each possible base on either side of the edited base. Information content was calculated by scaling each frequency by the log-ratio of the calculated base frequency to the background frequency (0.25). Information content was plotted as the height for each 5′ and 3′ context base to generate a Kullback–Leibler sequence logo[52]. Plots were created using Logomaker in Python[53].

## Energy modeling of CBE6 mutations
Mutations that arose during evolution were substituted into the ABE8e cryo-EM structure (PDB: 6VPC), and folding energies were computed to compare the stabilization of the monomeric and dimeric states of TadA*[34]. Structures with or without the substituted mutations were energy minimized in PyRosetta with the FastRelax protocol using the ref2015 energy function[54]. The difference in folding energy between each mutant and the original TadA* were calculated to estimate stabilization effects.

## HEK293T transfection and lysis
HEK293T cells (ATCC, CRL-3216) were acquired from ATCC and cultured in Dulbecco's Modified Eagle's Medium (DMEM) supplemented with GlutaMAX (Thermo Fisher Scientific) and 10% (v/v) Fetal Bovine Serum (FBS) (Gibco, qualified). The cells were incubated and cultured at 37 °C with 5% $CO_2$.

Prior to transfection, cells were seeded at a density of $1.6 \times 10^4$ cells per well in 96-well plates (Corning) and allowed to adhere for 16–24 hours. Cells were transfected when they reached approximately 60–80% confluency. For the transfection, 0.5 uL of Lipofectamine 2000 (Thermo Fisher Scientific) was combined with editor plasmid (100 ng) and guide RNA plasmid (40 ng), and the mixture was diluted into Opti-MEM reduced serum media (Thermo Fisher Scientific) to a final volume of 12.5 μl. Transfection was carried out according to the manufacturer's instructions.

After 72 hours, the culture media was removed, cells were washed with 100 μl of 1× PBS solution, and genomic DNA was extracted by adding 50 μl of lysis buffer per well. The lysis buffer contained 10 mM Tris-HCl (pH 8.0), 0.05% SDS, and 20 μg/ml of Proteinase K (New England Biolabs). The cell lysate was incubated at 37 °C for 1 hour, transferred to 96-well PCR plates, and heat-inactivated at 80 °C for 30 minutes. The genomic DNA was then stored at −20 °C.

## High-throughput sequencing
The genomic DNA from mammalian cell lines was subjected to high-throughput sequencing using a methodology outlined in a previous study[2]. The primer pairs employed in PCR 1 for all genomic sites are located in Supplementary Data 1. A 25 μl reaction for a given PCR 1 consisted of 0.125 uL of both forward and reverse primers, 1 μl of genomic DNA extract, 0.75 uL DMSO, 5 uL of Phusion Green HF Buffer (Thermo Fisher Scientific), 0.5 uL dNTPs, and 0.25 uL of Phusion Hot Start II DNA polymerase (Thermo Fisher Scientific). PCR1 reactions were conducted with the following parameters: an initial denaturation at 95 °C for 2 min, followed by 30 cycles of (95 °C for 10 s, 61 °C for 20 s, and 72 °C for 30 s), concluding with a final 72 °C extension for 5 min. In PCR 2, unique Illumina barcoding primer pairs were introduced. A 25 μl reaction for a given PCR 2 included 0.5 μM of each unique forward and reverse Illumina barcoding primer pair, 1 μl of unpurified PCR 1 reaction mixture, 5 uL of Phusion Green HF Buffer (Thermo Fisher Scientific), 0.5 uL dNTPs, and 0.25 uL of Phusion Hot Start II DNA polymerase (Thermo Fisher Scientific). The PCR2 reactions were conducted with the following parameters: an initial denaturation at 95 °C for 2 min, 10 cycles of (95 °C for 10 s, 61 °C for 20 s, and 72 °C for 30 s), and a final 72 °C extension for 5 min. Subsequently, the PCR products underwent purification through electrophoresis with a 1% agarose gel, utilizing a

QIAquick Gel Extraction Kit and eluting with 20 μl H2O. DNA concentrations were determined through a Qubit dsDNA High Sensitivity Assay Kit (Thermo Fisher Scientific). Subsequently, the samples were sequenced on an Illumina MiSeq instrument, and demultiplexing was carried out using the MiSeq Reporter software (Illumina). The resulting demultiplexed sequencing reads were subjected to analysis using CRISPResso2 and Microsoft Excel (version 16.75).

## DNA off-target editing analysis
Along with the editor plasmid (100 ng) and an SpCas9 gRNA plasmid (40 ng), a catalytically dead SaCas9 and an SaCas9 guide RNA plasmid were transfected into HEK293T cells and analyzed using high-throughput sequencing following the aforementioned procedure[26].

## RNA off-target editing analysis
The procedure for analyzing off-target RNA editing was conducted following established methods[26,41]. HEK293T cells were grown in two 96-well plates and subjected to parallel transfections with 250 ng of editor-encoding plasmids and 83 ng of *EMX1* guide RNA per well. After 48 hours, one plate was utilized to assess on-target DNA editing at the *EMX1* locus. For the second plate, cells were lysed using the RNeasy kit (Qiagen). After removing the medium, cells were washed with 1× PBS and lysed in RLT Plus Buffer (Qiagen). The lysate was then transferred to a DNA eliminator column and the flowthrough was treated with ethanol, which was then transferred to an RNeasy spin column. Samples underwent RW1 washing, followed by on-column DNA digestion using RNase-Free DNase in RDD buffer (Qiagen). Subsequent washes utilized RW1 and RPE buffers. Elution of RNA was done with 45 μl nuclease-free water, and each sample was supplemented with 2 μl of RNaseOUT (Thermo Fisher Scientific).

For cDNA synthesis, the SuperScript IV First-Strand Synthesis Kit (Thermo Fisher Scientific) was employed. RNA annealing with the OligodT primer occurred through heating at 65 °C, succeeded by cooling on ice for 1 minute. The resulting mixture underwent a reverse transcription reaction. Controls without reverse transcriptase were integrated to monitor genomic DNA contamination. Incubation was done at 50 °C for 10 minutes and 80 °C for 10 minutes, followed by cooling on ice for 1 minute. Optional RNA degradation with RNaseH was undertaken to enhance cDNA amplification efficiency. The first round of targeted amplicon sequencing PCR utilized 1 μl of each cDNA sample; subsequent sequencing steps were the same as the high-throughput, targeted genomic DNA sequencing method described above.

## Base editor mRNA synthesis from IVT
Production of base editor mRNA involved the generation of PCR products derived from a template plasmid harboring the expression construct for the desired base editor, a procedure outlined in prior work[10]. Amplification of the PCR product was performed in a total reaction volume of 200 μl, utilizing the IVT-F forward primer and IVT-R reverse primer. The resulting PCR product was purified using the QIAquick PCR Purification Kit (Qiagen) and subsequently eluted in 50 μl of nuclease-free water. In vitro transcription (IVT) reactions were initiated utilizing the HiScribe T7 High-Yield RNA Synthesis Kit (New England Biolabs), with a notable adaptation: N1-methyl-pseudouridine (substituting uridine) and co-transcriptional capping with CleanCap AG. Extraction of mRNA was performed through lithium chloride precipitation. For each 160 uL IVT reaction, 0.5 volumes of 7.5 M lithium chloride were introduced and thoroughly mixed. The mixture was incubated for 30 minutes at −20 °C, and a subsequent centrifugation step at 15,000 g for 20 minutes separated the supernatant from the pellet. Discarding the supernatant, the pellet was resuspended using 400 μl of ice-cold 70% ethanol. A second centrifugation, this time at 4 °C for 15 minutes, was performed, and the supernatant was discarded. The resulting pellet was air-dried at room temperature

for 5 minutes and was reconstituted in 100–200 μl of nuclease-free water. The samples were adjusted to a uniform concentration of 2 μg μl⁻¹ and conserved at a temperature of −80 °C.

## Nucleofection of patient-derived fibroblasts

Patient-derived fibroblasts were obtained from the Coriell Institute (GM03348) and cultured in DMEM with GlutaMAX supplemented with 15% FBS at 37 °C with 5% CO2. $2.0×10^5$ fibroblasts were nucleofected with 50 pmol of sgRNA (Synthego) and 1 μg of in vitro-transcribed SpCas9 mRNA via program DS-150 on a Lonza Nucleofector 4-D, which required 20 uL of P2 primary cell solution. Subsequently, cells were plated on 24-well plates, and the medium was changed after 24 h. After 72 h, the medium was removed, cells were washed with 1× PBS, and genomic DNA was extracted with 150 μl lysis buffer (10 mM Tris-HCl, pH 7.0, 0.05% SDS, 25 μg ml⁻¹ proteinase K).

## Statistics & reproducibility

Experiments were independently repeated three times unless otherwise stated. No data were excluded from analyses. The experiments were not randomized. The investigators were not blinded to allocation during experiments and outcome assessment. *P* values were calculated using Student's two-tailed, unpaired *t*-tests. *P* values of $< 0.05$ were considered statistically significant.

## Reporting summary

Further information on research design is available in the Nature Portfolio Reporting Summary linked to this article.

## Data availability

High-throughput DNA sequencing FASTQ files generated in this study have been deposited in the National Center of Biotechnology's Information Sequence Read Archive under BioProject "PRJNA1028129". Amino acid sequences of deaminases recommended in this work are listed in the Supplementary Information. The published structure of ABE8e is available in the Protein Data Bank (6VPC). Source data are provided with this paper.

## Code availability

All code used for processing library data is available on GitHub at https://github.com/MLE-zhang/BE_Lib.

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

## Acknowledgements
This work was supported by US National Institutes of Health (NIH) R35GM118062, U01AI142756, R01EB027793, R01EB031172, R01HL156647, U19NS132304, U19NS132315, and Howard Hughes Medical Institute (HHMI). We thank A. Sousa, S. Erwood, P. Randolph, S. DeCarlo, and S. Pandey for materials, discussion, and technical advice. M.E.N. was supported by a Ruth L. Kirschstein National Research Service Awards Postdoctoral Fellowship (GM143776-02). N.A.K. is a National Science Foundation (NSF) Graduate Research Fellow.

## Author contributions
E.Z. designed and cloned plasmids and phage, executed the evolution experiments, validated CBE editor activity in *E. coli* and mammalian cells, performed DNA off-target experiments, produced base editor mRNA and performed nucleofections in fibroblasts, and analyzed data. M.E.N. designed and cloned plasmids and phage, designed and advised the evolution experiments, developed and validated CBE editor activity for profiling in *E. coli*, and analyzed data. N.A.K. performed Cas-independent RNA off-target experiments, performed energy modeling, and analyzed *E. coli* library profiling data. E.Z., M.E.N., and D.R.L. designed the research. E.Z., M.E.N., and D.R.L. drafted the manuscript, with input from all authors.

## Competing interests
The authors declare competing financial interests: The Broad Institute has filed a patent application on behalf of E.Z., M.E.N., and D.R.L on the base editors developed in this study. D.R.L. is a consultant for Prime Medicine, Beam Therapeutics, Pairwise Plants, Chroma Medicine, and Nvelop Therapeutics, companies that use or deliver agents for genome editing, epigenome engineering, or PACE, and owns equity in these companies. The remaining authors declare no competing interests.
