## [Peer Review File · Nature Communications]

REVIEWER COMMENTS

Reviewer #1 (Remarks to the Author):

The manuscript by Liu et al. describes the phage-assisted evolution of next generation cytosine base editors (CBE6 variants) with increased C to T editing activity and decreased A to G editing relative to existing base editors. Current state-of-the-art TadA-derived cytosine base editors (TadCBEs) suffer from residual A to G editing and sequence-context dependent editing efficiencies. The approach is unique in that the CBE6 variants are evolved from an adenine and cytosine base editor (TadDE) rather than adenine base editors. The authors report the discovery of two key mutations for base editor performance: N46(I,V,L,or C) and Y73P. After characterizing the activity and specificity of CBE6 variants in *E. coli*, the authors compare them to existing CBEs in mammalian cells. The evolved CBE6 variants are reported to have higher editing efficiencies relative to existing base editors (>100-fold difference at protospacer position 6 in *E. coli* relative to TadCBE) and nearly no detectable A to G editing. However, the editing efficiency of CBE6a, one of the final variants, is sequence-context dependent. Further, the editing window of the CBE6 variants is larger than that of TadCBE. Overall, the presented CBE6 variants incrementally improve upon current cytosine base editors. Reductions in editing window size and off-target activity are necessary for safe therapeutic applications.

Major points

1. The title of the manuscript suggests that the CBE6 editors exhibit minimal sequence-context dependence. While almost no sequence-context preference is observed for CBE6b and CBE6c, CBE6a and CBE6d have comparable sequence context dependencies to that of TadCBE (Figure 2b). Given that the authors recommend the use of CBE6a for certain applications, the claim that sequence-context dependence is attenuated is somewhat misleading.
2. The CBE6 variants have an editing window spanning positions 4-8. This is far from ideal for many therapeutic applications, such as those requiring SNP editing. Other groups have developed CBEs with windows of 1-2 nucleotides (Kim et al., 2017). The authors should address this weakness.
3. The Cas-dependent off-target activities of the CBE6 variants is comparable to those of existing base editors. Reduced Cas-dependent off-target activity would substantially elevate the significance of this work. The reported reduction in Cas-independent off-target activity is not as relevant, as overall off-target activity is the limiting factor in the utility of base editors. The authors should provide more data or explanation on this important issue.
4. Some aspects of this work are not novel. The importance of the N46 locus in reducing A to G editing has been reported (Chen et al., 2016). Likewise, the V106W mutation has already been shown to reduce off-target activity.
5. The authors claim a significant increase in editing efficiency relative to existing technologies. Over a 100-fold difference is observed in *E. coli*; however, in mammalian cells, the CBE6s demonstrate only a

slight increase in editing frequency relative to existing technologies (Figures 3a and 3b). Thus, the CBE6 editors do not seem a significant advancement in base editing technology.

6. The figures and tables in the main text and supplementary figure lack analyses of statistical significance. This information is necessary to strengthen the arguments throughout the paper.

Minor points

1. In several figures, “A to I” and “C to U” are used, contrary to in the main text, where “A to G” and “C to T” are used. Consistent nomenclature would enhance clarity.
2. In many figures, y-axis scales are inconsistent. For instance, in Figure 3a, the range of the y-axis in the fourth plot (TRAC) differs from that of the other plots, which makes comparison of the loci difficult.

Reviewer #2 (Remarks to the Author):

The work presented by Zhang et al. focuses on a very relevant issue in the field of genome editing, namely increasing the specificity of cytosine base editors.

To this end, the authors describe a phage-assisted evolutionary selection system to design optimized CBEs that allow the generation of base editors with high C to T performance and almost undetectable A to T conversions. Importantly, the authors also confirmed the low Cas-dependent, Cas-independent and RNA-off target activity.

The authors also beautifully demonstrate that this novel CBE exhibited broad Cas domain compatibility, showing it as a promising strategy to increase the accuracy and applicability of CBE.

Overall, this study provides more efficient CBE tools with particularly relevant application for the generation of stop codon mutations.

Main Concerns:

1. Did the authors analyze the potential indels generated in the target loci?
2. Have the authors analyzed the potential translocations generated in the edited cells?

It would be relevant to show whether this happens in human cells or to discuss and comment on this possibility in the manuscript.

Reviewer #3 (Remarks to the Author):

The Liu group used phage-assisted (non)continuous evolution to improve cytosine base editor CBE6 to focus only on C-to-T editing with virtually no A-to-G editing that is common in cytosine base editors. Not only did they use evolution to increase specificity of their CBE6 but also to increase its deaminase activity, because cytosine editors tend to be less active than adenine base editors. The CBE6 editors are highly reliable and efficient, in particular, they applied these editors to installation of stop codons to decrease protein expression involved in disease in mammalian cells.

PACE and PANCE are run in bacterial cells. They applied their best bacterially evolved CBE6 derivatives to human cells to insert a stop codon in a cholesterol-expressing gene. This proof-of-concept application showed that their CBE6s worked with higher fidelity than previously generated TADCBE and TadDE.

The paper has some thoughtful analyses, such as bottom of p. 6 where they discuss how deaminase variants probably did not arise during PACE due to epistasis. Another good insight is at the end of the Results where they make recommendations of which CBE6 versions to use based on your needs. This could be the end of the paper, as the Discussion (conclusion) paragraph does not add to the paper. Could reduce the Discussion section, at a minimum.

A few minor comments below.

p. 2, line 62. Define AAV.

p. 5. They mention sequence-context preferences of base editors in Fig 2a. Unclear in 2a how they have 32 sequences.

p. 6. Lines 202-4. They have 1.3-fold efficiency with only N46I and 1.3-fold with both mutations. Not sure what they mean: how could both numbers be 1.3, so could be typo. Generally check the numbers. Another typo on p. 7, line 12 "Td-CBE_{max}": should this be Tad? Line 222, efficiencies of <0.1-0.1%, another typo. p. 9, line 287 should be "residual."

p. 7, middle. It appears that residual A-to-G activity is higher in human cells, perhaps a bit higher. In E.coli, they got <0.1% residual activity. But in human, looks like 0.1-1.2%. Can they explain the difference?

p. 8. They talk about the V106W variant that they found previously. Can they add a sentence about what they think the V106W does? It appears to generically assist in decreasing off-target activity.

p. 11, line 363. They mention S2060 cells as being tetracycline-resistant, but they should also be streptomycin-resistant, as well. Should clarify.

p. 12, line 381 should be Plaque assays, not Phage assays. They also then mention S2060 cells carrying pJC175e plasmid. These activity-independent cells were previously described by the lab as 2208 cells. Should clarify.

p. 13, line 406. They wrote a big section on plaque assays, but did not explain how to calculate titers. They should include this information.

p. 13 , line 417. MP's mutagenesis activity is suppressed by glucose, and this should be explained. They only mention glucose once here, but should also explain that, for example, MP cell stocks should be stored in glucose, as well. Glucose is an important point.

Jumi Shin (with help from student Maryam Ali, she understands confidentiality)

Dear Reviewers,

Thank you for reviewing our manuscript, “Phage-assisted evolution of highly active and exquisitely selective cytosine base editors with minimal sequence context preference”. In response to the referees’ comments, we have made revisions and additions throughout the manuscript and supporting information to address each comment. Below are our point-by-point responses (in blue) to all reviewers’ comments.

Reviewer #1 (Remarks to the Author):

The manuscript by Liu et al. describes the phage-assisted evolution of next generation cytosine base editors (CBE6 variants) with increased C to T editing activity and decreased A to G editing relative to existing base editors. Current state-of-the-art TadA-derived cytosine base editors (TadCBEs) suffer from residual A to G editing and sequence-context dependent editing efficiencies. The approach is unique in that the CBE6 variants are evolved from an adenine and cytosine base editor (TadDE) rather than adenine base editors. The authors report the discovery of two key mutations for base editor performance: N46(I,V,L,or C) and Y73P. After characterizing the activity and specificity of CBE6 variants in *E. coli*, the authors compare them to existing CBEs in mammalian cells. The evolved CBE6 variants are reported to have higher editing efficiencies relative to existing base editors (>100-fold difference at protospacer position 6 in *E. coli* relative to TadCBE) and nearly no detectable A to G editing. However, the editing efficiency of CBE6a, one of the final variants, is sequence-context dependent. Further, the editing window of the CBE6 variants is larger than that of TadCBE. Overall, the presented CBE6 variants incrementally improve upon current cytosine base editors. Reductions in editing window size and off-target activity are necessary for safe therapeutic applications.

We are thankful for the reviewer’s helpful comments and suggestions and hope that we have addressed their questions below.

Major points

1. The title of the manuscript suggests that the CBE6 editors exhibit minimal sequence-context dependence. While almost no sequence-context preference is observed for CBE6b and CBE6c, CBE6a and CBE6d have comparable sequence context dependencies to that of TadCBE (Figure 2b). Given that the authors recommend the use of CBE6a for certain applications, the claim that sequence-context dependence is attenuated is somewhat misleading.

We thank the reviewer for highlighting this point. We also recommend CBE6b, which is highly active, selective, and has minimal sequence context preference, and thus feel the title is accurate (within the limitations of the title size restrictions). That said, we also added clarifying language in the section discussing our recommendations to highlight the differences noted by the reviewer.

2. The CBE6 variants have an editing window spanning positions 4-8. This is far from ideal for many therapeutic applications, such as those requiring SNP editing. Other groups have developed CBEs with windows of 1-2 nucleotides (Kim et al., 2017). The authors should address this weakness.

We thank the reviewer for this suggestion. We have included a sentence commenting on the tradeoff of on-target editing activity and editing window size, as narrow window editors are generally less active. We acknowledge that trading activity for a narrower editing window may be appropriate for certain applications and have added this point in the text.

3. The Cas-dependent off-target activities of the CBE6 variants is comparable to those of existing base editors. Reduced Cas-dependent off-target activity would substantially elevate the significance of this work. The reported reduction in Cas-independent off-target activity is not as relevant, as overall off-target activity is the limiting factor in the utility of base editors. The authors should provide more data or explanation on this important issue.

We thank the reviewer for this comment. Cas-dependent off-target editing is more easily addressed than Cas-independent off-target editing since the former can be ameliorated by varying guide RNA sequence or length, PAM sequence targeted, and Cas domain used to target DNA. We have included a thorough characterization of the CBE6 variants for Cas-dependent off-target editing in SI Fig. 16-19. We have also tested our evolved deaminases with SpCas9 and eNme2C-Cas9, and are therefore optimistic that our evolved editor will also be compatible with other Cas domains, including those with reduced off-target engagement.

4. Some aspects of this work are not novel. The importance of the N46 locus in reducing A to G editing has been reported (Chen et al., 2016). Likewise, the V106W mutation has already been shown to reduce off-target activity.

We cited precedence for the N46 position in TadA when discussing the mutations (Chen et al., 2023). While we acknowledge that the previous report of the mutation at position N46 is important, in this work we focus on how the N46 mutation participates in additive effects with Y73P and other TadDE mutations. This is clear when we compared the CBE6 variants to Td-CBEmax, which is the editor identified in Chen et al., 2023 with the N46 mutation. As demonstrated in Fig. 3, Td-CBEmax has substantially less activity than the CBE6s as well as less activity than previous state-of-the-art CBEs such as BE4max and evoAPOBEC.

5. The authors claim a significant increase in editing efficiency relative to existing technologies. Over a 100-fold difference is observed in *E. coli*; however, in mammalian cells, the CBE6s demonstrate only a slight increase in editing frequency relative to existing technologies (Figures 3a and 3b). Thus, the CBE6 editors do not seem a significant advancement in base editing technology.

In Fig. 3b, we demonstrate that there is a substantial increase in the CBE6 editing levels compared to existing technologies when using non-SpCas9 targeting domains in mammalian cells, as demonstrated with eNme2C-Cas9 (CBE6d variants showed >10% higher average peak editing levels than any other CBE tested). Furthermore, a major advancement of the CBE6 editors compared to previously reported TadA-derived CBEs is the virtually undetected A•T-to-G•C editing.

6. The figures and tables in the main text and supplementary figure lack analyses of statistical significance. This information is necessary to strengthen the arguments throughout the paper.

We thank the reviewer for this suggestion. We have added statistical significance tests to support all key claims in the main text, which we have added as Supplementary Figures 23-31.

Minor points

1. In several figures, “A to I” and “C to U” are used, contrary to in the main text, where “A to G” and “C to T” are used. Consistent nomenclature would enhance clarity.

Thank you; we have corrected this.

2. In many figures, y-axis scales are inconsistent. For instance, in Figure 3a, the range of the y-axis in the

fourth plot (TRAC) differs from that of the other plots, which makes comparison of the loci difficult.

We thank the reviewer for pointing this out. This was intentional because scaling the y-axis can make the bars for some sites hard to see. Therefore, we elected to prioritize scaling the y-axis such that it is easier to compare *between editors* at a given site. We have made sure to clearly label the y-axes so that the readers can more readily see that they are different.

Reviewer #2 (Remarks to the Author):

The work presented by Zhang et al. focuses on a very relevant issue in the field of genome editing, namely increasing the specificity of cytosine base editors. To this end, the authors describe a phage-assisted evolutionary selection system to design optimized CBEs that allow the generation of base editors with high C to T performance and almost undetectable A to T conversions. Importantly, the authors also confirmed the low Cas-dependent, Cas-independent and RNA-off target activity. The authors also beautifully demonstrate that this novel CBE exhibited broad Cas domain compatibility, showing it as a promising strategy to increase the accuracy and applicability of CBE. Overall, this study provides more efficient CBE tools with particularly relevant application for the generation of stop codon mutations.

We are grateful for the reviewer's kind words and have hopefully addressed their concerns below.

Main Concerns:

1. Did the authors analyze the potential indels generated in the target loci?

In SI Fig. 14-15, we have included indel levels for all SpCas9 and eNme2C-Cas9 sites tested.

2. Have the authors analyzed the potential translocations generated in the edited cells? It would be relevant to show whether this happens in human cells or to discuss and comment on this possibility in the manuscript.

We thank the reviewer for this suggestion and have added a discussion of this possibility in the main text. Webber et al., 2019 report that translocations were virtually undetectable via ddPCR using base editing with optimal reagents. Fiumara et al., 2023 reported that potential translocations generated by base editors are correlated with the fraction of indels detected. As shown in SI Fig. 14-15, CBE6 indel levels are low and comparable to previously reported CBEs and ABEs, which suggests that the CBE6 variants will not generate more translocations than current state-of-the-art editors. We have updated the main text to reflect these considerations.

Reviewer #3 (Remarks to the Author):

The Liu group used phage-assisted (non)continuous evolution to improve cytosine base editor CBE6 to focus only on C-to-T editing with virtually no A-to-G editing that is common in cytosine base editors. Not only did they use evolution to increase specificity of their CBE6 but also to increase its deaminase activity, because cytosine editors tend to be less active than adenine base editors. The CBE6 editors are highly reliable and efficient, in particular, they applied these editors to installation of stop codons to decrease protein expression involved in disease in mammalian cells.

PACE and PANCE are run in bacterial cells. They applied their best bacterially evolved CBE6 derivatives to human cells to insert a stop codon in a cholesterol-expressing gene. This proof-of-concept application showed that their CBE6s worked with higher fidelity than previously generated TADCBE and TadDE.

The paper has some thoughtful analyses, such as bottom of p. 6 where they discuss how deaminase variants probably did not arise during PACE due to epistasis. Another good insight is at the end of the Results where they make recommendations of which CBE6 versions to use based on your needs. This

could be the end of the paper, as the Discussion (conclusion) paragraph does not add to the paper. Could reduce the Discussion section, at a minimum.

We appreciate the reviewer's encouraging comments.

A few minor comments below.

p. 2, line 62. Define AAV.

Thank you; we have clarified this.

p. 5. They mention sequence-context preferences of base editors in Fig 2a. Unclear in 2a how they have 32 sequences.

We apologize for the lack of clarity. In Fig. 2a, the library is constructed with the target C or A at position 6 in the protospacer, with all possible flanking nucleotides before the target base (4 total) and after the target base (4 total). This leads to 16 possible combinations for the target C and 16 possible combinations for the target A, leading to 32 total sequences. We have added more details in the main text to improve clarity.

p. 6. Lines 202-4. They have 1.3-fold efficiency with only N46I and 1.3-fold with both mutations. Not sure what they mean: how could both numbers be 1.3, so could be typo. Generally check the numbers.

We have checked our calculations, and the numbers are correct. In this experiment, the addition of Y73P to TadCBE_d N46I does not have an effect on the editing efficiency at position 6 of the protospacer using the *E. coli* library.

Another typo on p. 7, line 12 "Td-CBEmax": should this be Tad?

Chen et al., 2023 reported Td-CBEmax, spelled as is.

Line 222, efficiencies of <0.1-0.1%, another typo.

As the limit of detection of HTS is 0.1%, we found residual A•T-to-G•C editing below the limit of detection or at 0.1%.

p. 9, line 287 should be "residual."

Thank you; we have corrected this error.

p. 7, middle. It appears that residual A-to-G activity is higher in human cells, perhaps a bit higher. In *E. coli*, they got <0.1% residual activity. But in human, looks like 0.1-1.2%. Can they explain the difference?

Deoxyinosines can be repaired to protect the genome in *E. coli* and mammalian cells through base excision repair (BER) and alternative excision repair (AER) pathways (Kuraoka et al., 2015). Lesion recognition and base removal through BER is primarily accomplished by alkyl-adenine DNA glycosylases (AlkA in *E. coli* and AAG in *H. sapiens*), which could differ in their efficiencies across organisms (Kuraoka et al., 2015). Additionally, an AER pathway using endonuclease V for the removal of deoxyinosine from DNA has been identified in *E. coli*, but an analogous pathway has not been found in mammalian cells (Kuraoka et al., 2015). We hypothesize that the differences in these repair pathways for

inosine when comparing *E. coli* and mammalian cells may explain the difference between the residual A•T-to-G•C activity we see. We have now added an explanation in the text.

p. 8. They talk about the V106W variant that they found previously. Can they add a sentence about what they think the V106W does? It appears to generically assist in decreasing off-target activity.

V106W was initially reported during efforts to reduce off-target RNA editing by decreasing deaminase interaction with RNA by steric occlusion (Rees et al., 2019). V106W also decreases off-target editing of DNA, perhaps by a similar mechanism (SI Fig. 16-22). However, on-target activity on DNA is largely preserved, possibly due to the high local concentration of the DNA substrate that is enforced by fusion to Cas9. We have added this explanation in the main text.

p. 11, line 363. They mention S2060 cells as being tetracycline-resistant, but they should also be streptomycin-resistant, as well. Should clarify.

Thank you; we have clarified this statement in the revised manuscript.

p. 12, line 381 should be Plaque assays, not Phage assays. They also then mention S2060 cells carrying pJC175e plasmid. These activity-independent cells were previously described by the lab as 2208 cells. Should clarify.

Thank you; we have clarified this statement in the revised manuscript.

p. 13, line 406. They wrote a big section on plaque assays, but did not explain how to calculate titers. They should include this information.

Thank you; we have included this information in the revised manuscript.

p. 13, line 417. MP's mutagenesis activity is suppressed by glucose, and this should be explained. They only mention glucose once here, but should also explain that, for example, MP cell stocks should be stored in glucose, as well. Glucose is an important point.

Thank you; we have included this information in the revised manuscript.

We appreciate your helpful suggestions on our manuscript and hope that our revisions address your concerns. The revised manuscript has been significantly strengthened, and we are hopeful that this work will be an impactful contribution to *Nature Communications*.

REVIEWERS' COMMENTS

Reviewer #1 (Remarks to the Author):

The revised manuscript describes the phage-assisted evolution of next generation cytosine base editors (CBE6 variants) with increased C●G to T●A editing activity and decreased A●T to G●C editing relative to existing base editors. The authors have made some changes to the main text that clarify the limited utility of their base editors given the sequence-context editing efficiency of CBE6a and the larger editing window of the CBE6 variants relative to other base editors. These comments soften the previous claims of minimal sequence-context dependence and general superiority of the CBE6 variants relative to existing base editors. In addition, the authors added analyses of statistical significance in the supplemental material of their major claims, which strengthens this work. However, no new data is added, and the technology still represents an incremental improvement over current base editors. Aside from a significant reduction in A●T to G●C editing, the CBE6 variants perform similarly to existing technologies.

Major points

1. The authors claim a significant increase in editing efficiency relative to existing technologies. Over a 100-fold difference is observed in *E. coli*; however, in mammalian cells, the CBE6s demonstrate a slight increase in editing frequency relative to existing technologies (Figures 3a and 3b). Thus, the CBE6 editors are not a significant advancement in base editing technology.
2. The Cas-dependent off-target activities of the CBE6 variants is comparable to those of existing base editors. Reduced Cas-dependent off-target activity would substantially elevate the significance of this work. The reported reduction in Cas-independent off-target activity is not as relevant, as overall off-target activity is the limiting factor in the utility of base editors. The authors addressed this well in the comments to the reviewer, but inclusion of their explanation in the text is desirable.

Reviewer #2 (Remarks to the Author):

The authors have addressed my concerns and have included new information in the paper that improves the quality of the paper,.

In my opinion this study is ready for publication

Reviewer #3 (Remarks to the Author):

The authors addressed all our points, and the paper is a good addition to Nature Communications.

Jumi Shin and Maryam Ali

February 6, 2024

Dear Reviewers,

Thank you for reviewing our manuscript, “Phage-assisted evolution of highly active cytosine base editors with enhanced selectivity and minimal sequence context preference”. In response to the referees’ comments, we have made revisions and additions throughout the manuscript and supporting information to address each comment. Below are our point-by-point responses (in blue) to all reviewers’ comments.

Reviewer #1 (Remarks to the Author):

The revised manuscript describes the phage-assisted evolution of next generation cytosine base editors (CBE6 variants) with increased C●G to T●A editing activity and decreased A●T to G●C editing relative to existing base editors. The authors have made some changes to the main text that clarify the limited utility of their base editors given the sequence-context editing efficiency of CBE6a and the larger editing window of the CBE6 variants relative to other base editors. These comments soften the previous claims of minimal sequence-context dependence and general superiority of the CBE6 variants relative to existing base editors. In addition, the authors added analyses of statistical significance in the supplemental material of their major claims, which strengthens this work. However, no new data is added, and the technology still represents an incremental improvement over current base editors. Aside from a significant reduction in A●T to G●C editing, the CBE6 variants perform similarly to existing technologies.

We are thankful for the reviewer’s helpful comments and suggestions and hope that we have addressed their comments below.

Major points

1. The authors claim a significant increase in editing efficiency relative to existing technologies. Over a 100-fold difference is observed in *E. coli*; however, in mammalian cells, the CBE6s demonstrate a slight increase in editing frequency relative to existing technologies (Figures 3a and 3b). Thus, the CBE6 editors are not a significant advancement in base editing technology.

In Fig. 3b, we demonstrate that there is a substantial increase in the CBE6 editing levels compared to existing technologies when using non-SpCas9 targeting domains in mammalian cells, as demonstrated with eNme2C-Cas9 (CBE6d variants showed >10% higher average peak editing levels than any other CBE tested). These data demonstrate that the CBE6 variants can substantially advance cytosine base editing using smaller Cas9 domains, expanding the genome-targeting scope with alternative PAMs and is especially helpful for compatibility with delivery technologies.

TadCBEs are described as highly active in Neugebauer et al. (2023), so we are comfortable describing CBE6 variants as highly active since they perform comparably or better than TadCBEs across all sites tested. As all of our data to date suggest that CBE6s are always comparable to or better than current CBE technologies, they should be useful in many applications that prioritize editing efficiency.

Furthermore, a major advancement of the CBE6 editors compared to previously reported TadA-derived CBEs is the virtually undetected A●T-to-G●C. For example, CBE6 variants showed up to 346-fold higher selectivity for C-to-T over A-to-G editing than TadCBE in human cells.

2. The Cas-dependent off-target activities of the CBE6 variants is comparable to those of existing base editors. Reduced Cas-dependent off-target activity would substantially elevate the significance of this work. The reported reduction in Cas-independent off-target activity is not as relevant, as overall off-target

activity is the limiting factor in the utility of base editors. The authors addressed this well in the comments to the reviewer, but inclusion of their explanation in the text is desirable.

Thank you; we have now included the explanation in the main text.

We appreciate your helpful suggestions on our manuscript and hope that our revisions address your concerns. The revised manuscript has been significantly strengthened, and we are hopeful that this work will be an impactful contribution to *Nature Communications*.